# Nc*GRA7* and Nc*ROP40* Play a Role in the Virulence of *Neospora caninum* in a Pregnant Mouse Model

**DOI:** 10.3390/pathogens11090998

**Published:** 2022-08-31

**Authors:** Laura Rico-San Román, Rafael Amieva, Javier Regidor-Cerrillo, Marta García-Sánchez, Esther Collantes-Fernández, Iván Pastor-Fernández, Jeroen P. J. Saeij, Luis Miguel Ortega-Mora, Pilar Horcajo

**Affiliations:** 1SALUVET, Animal Health Department, Faculty of Veterinary Sciences, Complutense University of Madrid, Ciudad Universitaria s/n, 28040 Madrid, Spain; 2SALUVET-Innova S.L., Faculty of Veterinary Sciences, Complutense University of Madrid, Ciudad Universitaria s/n, 28040 Madrid, Spain; 3Department of Pathology, Microbiology and Immunology, School of Veterinary Medicine, University of California, Davis, CA 95616, USA

**Keywords:** *Neospora caninum*, NcGRA7 protein, NcROP40 protein, CRISPR/Cas9, BALB/c, pregnant mouse model, virulence factor

## Abstract

The intraspecific variability among *Neospora caninum* isolates in their in vitro behaviour and in vivo virulence has been widely studied. In particular, transcriptomic and proteomic analyses have shown a higher expression/abundance of specific genes/proteins in high-virulence isolates. Consequently, the dense granule protein NcGRA7 and the rhoptry protein NcROP40 were proposed as potential virulence factors. The objective of this study was to characterize the role of these proteins using CRISPR/Cas9 knockout (KO) parasites in a well-established pregnant BALB/c mouse model of *N. caninum* infection at midgestation. The deletion of Nc*GRA7* and Nc*ROP40* was associated with a reduction of virulence, as infected dams displayed milder clinical signs, lower parasite burdens in the brain, and reduced mortality rates compared to those infected with the wild-type parasite (Nc-Spain7). Specifically, those infected with the Nc*GRA7* KO parasites displayed significantly milder clinical signs and a lower brain parasite burden. The median survival time of the pups from dams infected with the two KO parasites was significantly increased, but differences in neonatal mortality rates were not detected. Overall, the present study indicates that the disruption of Nc*GRA7* considerably impairs virulence in mice, while the impact of Nc*ROP40* deletion was more modest. Further research is needed to understand the role of these virulence factors during *N. caninum* infection.

## 1. Introduction

*Neospora caninum* is an obligate intracellular apicomplexan parasite, phylogenetically related to *Toxoplasma gondii*, and the aetiological agent of bovine neosporosis. This disease is one of the major causes of reproductive failure in cattle worldwide [1], resulting in significant economic losses [2]. Unfortunately, no tools are available for the treatment or prophylactic control of bovine neosporosis [1].

The intraspecific biological variability of *N. caninum* has been widely demonstrated. *N. caninum* isolates have demonstrated differences in parasite growth in cell culture [3,4,5] as well as differences in pathogenicity and vertical transmission in pregnant models in mice [3,6,7] and cattle [8,9,10,11]. A positive correlation exists between certain in vitro phenotypes and in vivo virulence [4]. However, the molecular factors underlying this variability remain unknown.

Studies of *N. caninum* population genomics have shown gene copy number variation among different isolates, but with limited sequence polymorphisms, and a global selective sweep of a single lineage [12]. Moreover, mutational hotspots have been identified, but the current annotation of the *N. caninum* genome is incomplete, which prevents assigning functional significance to these hotspots [13]. In contrast to *T. gondii*, little diversity has been detected at potential antigenic loci [14,15,16]. In recent years, comparisons between isolates with different virulence by omic technologies have highlighted differences in proteomic and transcriptomic profiles between isolates [17,18,19,20,21,22]. These differences provide a list of genes that can be used as a starting point for identifying factors that may have a key role in explaining such variability. Among these, dense granule NcGRA7 and rhoptry NcROP40 proteins were highlighted as putative virulence factors due to their higher expression and abundance in high-virulence isolates in these comparisons.

Rhoptry and dense granules are specialized secretory organelles unique to the phylum Apicomplexa [23] and are linked to parasite cell invasion, intracellular development, and host response control [24,25,26,27]. NcGRA7 is an immunodominant protein in *N. caninum* [28] that seems to be involved in cell invasion mechanisms [29,30]. In addition, recent studies in murine models have shown a partial decrease in the virulence of a mutant deficient in this protein [31,32,33]. NcROP40 has been less studied, but it may also be involved in the invasion of the host cell [34]. Furthermore, other rhoptry proteins, such as NcROP16 and NcROP5, have been described as virulence factors in mice [35,36].

The use of genetic tools for DNA manipulation in *N. caninum* will help us to unravel the function of specific genes in host–parasite interactions and pathogenesis. The goal of this study was to characterize the role of NcGRA7 and NcROP40 proteins through experimental infections in a well-established BALB/c mouse model for congenital and cerebral neosporosis [37,38] using CRISPR/Cas9 knockout (KO) parasites for the two genes.

## 2. Materials and Methods

### 2.1. Generation of Knockout Parasites and Complemented Strains

Nc*ROP40* was disrupted using the CRISPR/Cas9 system as previously described [39]. Briefly, specific gRNA sequences targeting the 5′ and 3′ regions of the Nc*ROP40* (ToxoDB ID NCLIV_012920) coding sequence (Table 1) were cloned into the *BsaI* sites of the pSS013-Cas9 vector (pU6 plasmid, Addgene plasmid #52694) [40]. The pLoxP-mCherry-DHFR plasmid (Addgene Plasmid #70147), which contains the *Toxoplasma* DHFR–TS pyrimethamine-resistant allele marker, was used as a donor template (Figure 1A).

Approximately 1–2 × 10^7^ Nc-Spain7 tachyzoites were cotransfected with the two plasmids containing the gRNAs mentioned above, together with the *NotI*-linearized mCherry-DHFR–TS plasmid, at a final 5:1 molar ratio as previously described [39]. Twenty-four hours after transfection, parasite selection was carried out with 10 μM pyrimethamine (Sigma-Aldrich, St. Louis, MO, USA) for three passages. Subsequently, individual clones were isolated by limiting dilution in 96-well plates and confirmed using PCR by means of the observation of correct DHFR–TS integration into the Nc*ROP40* locus (Figure 1A, Table 1). Nc*ROP40*-deficient clones (NcΔ*ROP40*) were also confirmed by Western blot and immunofluorescence assays (see Section 2.1.2 and Section 2.1.3). The Nc*GRA7*-deficient strain (NcΔ*GRA7*) was generated previously [39]. In both cases, the high-virulence isolate Nc-Spain7 was used as the parental strain. This isolate has a controlled number of passages in vitro and has been widely characterized in both in vitro and in vivo models, which makes it an excellent reference isolate.

To generate complemented strains, exogenous copies of the respective genes were inserted into the uracil phosphoribosyl-transferase (UPRT) locus (NCLIV_056020) (Figure 1B). Since the UTR regions are not annotated for these genes on ToxoDB, the repair templates were constructed by amplifying the coding sequences including 1000 bp upstream of the start codon and downstream of the stop codon (Table 1). These amplicons were subsequently inserted into the multiple cloning site of the universal pUC19 plasmid, generating the pUC19-NcROP40 and pUC19-NcGRA7 plasmids. Transfections were performed as above with minor modifications. The pU6 plasmid containing the gRNA sequence targeting the 5′ end of the UPRT coding sequence and the *SbfI*-linearized pUC19-NcROP40 plasmid or the *EcoRI*-HF-linearized pUC19-NcGRA7 plasmid were cotransfected at a 1:5 gRNA:insert molar ratio. Complemented parasites were selected with 15 μM 5-fluorodeoxyuridine (FUDR, Sigma-Aldrich, St. Louis, MO, USA) for three passages, and single clones were obtained by limiting dilution as above. Finally, clones were analysed by PCR, Western blotting, and immunofluorescence to confirm the presence and expression of Nc*ROP40* and Nc*GRA7*. All the primers and gRNA sequences used are listed in Table 1.

#### 2.1.1. PCR

All PCR reactions were performed using Taq DNA polymerase (Ecogen, Madrid, Spain) in a 25 μL reaction mixture containing 5 μL of DNA as a template, following the manufacturer’s recommendations. For diagnostic PCRs, DNA from single clones was extracted using the Maxwell^®^ 16 Cell LEV DNA Purification Kit (Promega, Madison, WI, USA). A schematic representation of the diagnostic PCRs and primers used can be found in Figure 1A,B and Table 1.

#### 2.1.2. Western Blot

To assess NcROP40 and NcGRA7 expression, wild-type (WT, Nc-Spain7 isolate) and genetically modified strains (KOs and complemented strains) were analysed by Western blot as previously described [34]. Purified tachyzoites were disrupted by bath sonication, electrophoresed in 15% acrylamide gels under reducing conditions, and transferred onto nitrocellulose membranes (GE Healthcare, Chicago, IL, USA; 1.5 × 10^6^ tachyzoites per well). The antigen-coated membrane was cut into strips, and then the strips were blocked with TBS-Tween 20 buffer containing 5% (*w*/*v*) skimmed milk powder and incubated with polyclonal antibody (PAb) α-NcROP40 and α-NcGRA7 at a 1:1000 dilution. After three washes with TBS-Tween 20, goat anti-rabbit IgG monoclonal antibody conjugated to horseradish peroxidase (Sigma-Aldrich, St. Louis, MO, USA) was used as a secondary antibody at a 1:10,000 dilution. The strips were washed again, and reactions were developed using Immobilon Western chemiluminescent HRP (Bio-Rad, Hercules, CA, USA) as substrate until signal visualization. Image capturing was performed using a ChemiDoc XRS+ System (Bio-Rad, Hercules, CA, USA).

#### 2.1.3. Immunostaining of NcROP40 and NcGRA7

Immunofluorescence imaging was performed following a previously described protocol [34]. The supernatants of the cell cultures were discarded at 48 h postinfection (pi), and wells were washed three times with PBS and subsequently fixed with paraformaldehyde 3%–glutaraldehyde 0.05% or ice-cold methanol for NcGRA7 or NcROP40 staining, respectively. Wells were washed with PBS and cells were blocked and permeabilized with 300 μL/well of 3% BSA and 0.25% Triton-X 100 in PBS for 45 min at 37 °C, followed by additional washes with PBS. Then, the cultures were labelled with the monoclonal antibody (MAb) α-NcSAG1 (dilution 1:250) as a surface marker [41] and affinity purified PAb α-NcROP40 (dilution of 1:6) [34] and α-NcGRA7 (dilution of 1:6) [42] by incubation for 1 h at 37 °C. Affinity purified antibodies were prepared from PAbs following standard procedures [28]. After three additional washes with PBS, the cultures were incubated with Alexa Fluor 594-conjugated goat anti-mouse IgG and Alexa Fluor 488-conjugated goat anti-rabbit IgG (Life Technologies, Carlsbad, CA, USA) as secondary antibodies at a 1:1000 dilution for 1 h at room temperature in the dark and washed three times with PBS. In the final wash, DAPI stain was included to label the nuclei. Finally, the plates were washed with distilled water, and the protein location was visualized using an inverted fluorescence microscope (Nikon Eclipse TE200) at 100× magnification.

### 2.2. Parasite Culture and Inoculum Preparation

The parasites used in this study were routinely maintained by continuous passage in a monolayer culture of the MARC-145 cell line and incubated at 37 °C in a 5% CO_2_ humified incubator. Cells were cultured in DMEM (Sigma-Aldrich, St. Louis, MO, USA) supplemented with 10% heat-inactivated foetal calf serum (FCS; Gibco BRL, Thermo Fisher Scientific, Paisley, UK) and a mixture of penicillin (100 U/mL), streptomycin (100 μg/mL), and amphotericin B (Lonza Group, Basel, Switzerland).

Tachyzoites were harvested from culture flasks 3 days pi, when the majority were still intracellular, using a cell scraper and followed by a single passage through a 25-gauge needle. Tachyzoite viability and concentration were determined by trypan blue vital exclusion staining followed by counting in a Neubauer chamber. For the in vivo virulence assays, tachyzoites were adjusted with PBS to the required dose of 10^5^ tachyzoites in a final volume of 200 μL per mouse and were inoculated within 30 min of their collection [38].

### 2.3. Mice and Ethics Statement

A total of 80 female and 48 male 7-week-old BALB/c mice were purchased from Janvier Labs (Laval, France). The animals were free from common viral, parasitic, and bacterial pathogens according to the results of routine screening analyses performed by the manufacturer. Mice were housed with ad libitum access to food and water in a controlled environment with 12-h light and 12-h dark cycles. Animals were used for experimentation after an acclimatization period of at least 15 days.

Animal procedures were approved by the Animal Welfare and Experimentation Committee of the Complutense University of Madrid and the Animal Protection Area of the Community of Madrid, Spain (PROEX 274/16 and 66.7/20) and were performed according to the corresponding guidelines and the Spanish and UE legislation (Law 6/2013; Royal Decree 118/2021; Directive 2010/63/UE). The use of genetically modified organisms was approved by the Genetically Modified Organisms Committee, and its manipulation was adjusted to that described in the current legislation (Law 9/2003; Royal Decree 178/2004; Directive 2009/41/UE). All animals used in this study were handled in strict accordance with practices made to minimize suffering. As a humane endpoint, severely clinically affected animals (evident loss of body condition or nervous signs) were euthanized to limit unnecessary suffering.

### 2.4. Assays of Parasite Virulence in Mice

Two different experiments were carried out to compare the virulence of the NcΔ*GRA7* and NcΔ*ROP40* strains with the parental Nc-Spain7 isolate, as described below. In Experiment 1, well-established models for congenital and cerebral neosporosis were used as previously described [38,43]. In Experiment 2, the cellular immune response at the acute stage of infection was determined.

• Experiment 1

In this experiment, female mice were oestrus-synchronized by the Whitten effect and were mated for 96 h by housing one male with two females. Day 0 of pregnancy was defined as the first day that the females were housed with males. Female mice were then randomly distributed in groups of 20 mice each. At midgestation (Day 7 of gestation), mice were subcutaneously challenged with 10^5^ tachyzoites/mouse of the KO parasites (NcΔ*GRA7* or NcΔ*ROP40*), with 10^5^ tachyzoites/mouse of the complemented parasites (iNcΔ*GRA7* and iNcΔ*ROP40*), with 10^5^ tachyzoites/mouse of the WT isolate or with 200 μL of sterile PBS (negative control group). Pregnancy was confirmed by weighing on Days 15–18 of gestation, and pregnant mice were allocated individually for parturition. Female mice that did not become pregnant were housed in groups of up to 12 mice. Nonpregnant mice, dams, and their offspring were evaluated daily for clinical signs compatible with neosporosis and mortality. Clinical signs were scored according to the description from Pastor-Fernández et al. [43]. Briefly, depending on the severity of the clinical signs, the following scores were given: no alterations (score = 0), ruffled coat/rough hair coats (score = 1), rounded back (score = 2), severe weight loss (score = 3), and nervous signs (score = 4). Dams and offspring were euthanized in a CO_2_ chamber at 30 days postpartum (pp), while nonpregnant mice were euthanized at 30 days pi. Serum and brain samples from animals that were euthanized or reached the human end-point were collected and stored at −80 °C.

For the *N. caninum* congenital model, data on fertility rate, litter size, early pup mortality, and neonatal mortality were collected. The fertility rate was defined as the percentage of females that became pregnant. The litter size was defined as the number of pups delivered per dam. Stillbirth was evaluated as the number of full-term dead pups at birth, and early pup mortality was defined as the number of dead pups in the first two days pp. Neonatal mortality was defined as the number of dead pups from Day 2 to Day 30 pp. The model for cerebral neosporosis was performed with the dams and females that did not become pregnant at the chronic infection stage (Days 30 pp and 30 pi, respectively) by determining the presence and parasite burden of *N. caninum* DNA in the brain.

• Experiment 2

In order to assess the cellular immune responses triggered at the acute stage of infection, male mice were randomly distributed into four groups of five mice each and were subcutaneously challenged with 10^5^ tachyzoites/mouse of the WT isolate, with 10^5^ tachyzoites/mouse of the NcΔ*GRA7* strain, with 10^5^ tachyzoites/mouse of the NcΔ*ROP40* strain or with 200 μL of sterile PBS (negative control group). All animals were monitored daily for 4 weeks. Mice were sacrificed at 5 days pi, and spleen samples were collected and frozen at −80 °C to determine the expression levels of the proinflammatory cytokines interferon γ (IFN-γ) and tumour necrosis factor α (TNF-α) and the anti-inflammatory/regulatory cytokines interleukin 4 (IL-4) and interleukin 10 (IL-10) by quantitative real time PCR (RT-qPCR).

#### 2.4.1. Parasite Detection and Quantification

DNA extraction was carried out from 40 to 100 mg of brain tissue samples using the Maxwell^®^ 16 Mouse Tail DNA Purification Kit (Promega, Madison, WI, USA) following the manufacturer’s instructions. The concentrations of DNA were determined by spectrophotometry using a nanophotometer (NanoPhotomer^®^, Implen GmbH, Munich, Germany) and adjusted to 20 ng/μL with molecular-grade water.

Parasite detection and quantification were performed from approximately 100 ng of extracted DNA by real-time PCR using the 7500 Real-Time PCR System (Applied Biosystems, Foster City, CA, USA) and the commercial kit GoTaq^®^ qPCR Master Mix (Promega, Madison, WI, USA). Primers targeted the Nc-5 region to quantify parasites and the 28S rRNA gene to quantify host DNA, as previously described [44]. Parasite burden was calculated by the interpolation of cycle threshold (Ct) values on a standard curve. The standard curve was designed from the DNA equivalent to 10^−1^–10^5^ tachyzoites. To normalize the quantification of the parasite in each sample, a 28S standard curve was also designed (from 320 ng to 10 ng DNA). The results were expressed as the relation between the amounts of parasite DNA (number of tachyzoites) and host DNA (100 ng DNA).

#### 2.4.2. Real-Time RT–PCR of Cytokine Expression

RNA was extracted from approximately 10 mg of each spleen using a commercial Maxwell^®^ 16 LEV simplyRNA Purification Kit (Promega, Madison, WI, USA) following the manufacturer’s recommendations. RNA integrity was analysed using 1% agarose gel electrophoresis, and RNA concentrations were determined using a NanoPhotometer^®^ spectrophotometer (Implen GmbH, Munich, Germany). Reverse transcription was performed with the master mix SuperScript^®^ VILO™ cDNA Synthesis Kit (Invitrogen, Paisley, UK) in 20 μL reactions using up to 2.5 μg of total RNA for the synthesis of cDNA. The obtained cDNA products were diluted 1:20 in molecular-grade water and then diluted 3 times in 4-fold serial dilutions (1:80; 1:320; 1:1280) and were used in qPCR assays. Real-time PCR was performed using the 7500 Fast Real-Time PCR System (Applied Biosystems, Waltham, MA, USA) with the commercial kit Power SYBR^®^ Green PCR Master Mix (Applied Biosystems, Waltham, MA, USA) following the manufacturer’s instructions. Briefly, PCR reactions were conducted using 12.5 µL of Power SYBR^®^ Green PCR Master Mix (Applied Biosystems, Waltham, MA, USA), 10 pmol of each primer and 5 μL of diluted cDNA samples. The primer sequences for cDNA amplification of IFN-γ, TNF-α, IL-4, IL-10 and β-actin have been previously published [45,46] (Appendix A). All Ct values were normalized to the expression of the housekeeping gene β-actin, which exhibited comparable Ct values for all the samples. For relative quantification of gene expression, the comparative threshold cycle method was used. The relative n-fold change of each target cytokine expression, normalized to the endogenous reference and relative to the control group, is given by 2^−∆∆Ct^ [47]. In addition, the ratio of IFN-γ/IL-10 was calculated as an indicator of Th1/Th2 response modulation.

#### 2.4.3. Humoral Immune Responses

Serum levels of *N. caninum*-specific IgG1 and IgG2a isotypes were determined using ELISA in pregnant and nonpregnant mice (Experiment 1) as previously described [38]. The 96-well plates were coated with *N. caninum* soluble tachyzoite antigen (0.125 μg in 100 μL/well), and ELISA was performed using a 1:100 dilution of serum samples and an anti-mouse IgG1 or IgG2a peroxidase-conjugated secondary antibody (1:5000; Southern Biotechnology, Birmingham, AL, USA). The absorbance was measured at 405 nm, and the results were expressed as a relative index percentage (RIPC) using the following formula: RIPC = (OD sample − OD negative control)/(OD positive control − OD negative control) × 100, where OD is the mean value of the optical density. The antibody isotype balance was evaluated using the IgG1/IgG2a ratio.

### 2.5. Statistical Analysis

All statistical analyses were carried out using GraphPad Prism v.7.0 software (San Diego, CA, USA). Differences in data on fertility and mortality rates were evaluated using the chi-square test (χ^2^) or Fisher’s F test. Neonatal mortality was analysed using the Kaplan–Meier survival method to estimate the percentage of surviving animals at each time point. The log-rank (Mantel–Cox) test was applied to compare the survival curves between different groups and to calculate the median survival time. After testing the samples for normal distribution with D’Agostino–Pearson test, a one-way ANOVA followed by Tukey’s multiple range test was used to compare litter size and anti-*N. caninum* antibody levels. However, differences in clinical sign scores, parasite burden and cytokine expression levels between groups were analysed using the Kruskal–Wallis test followed by Dunn’s multiple-comparison test. Finally, to evaluate whether parasite burden correlated with clinical sign scores, regression analysis was performed to determine the Pearson correlation coefficient, r, and the strength of the relationship was expressed as r^2^. Statistical significance for all analyses was established at *p* < 0.05.

## 3. Results

### 3.1. Successful Construction of the NcROP40 Knockout and NcROP40 and NcGRA7 Complemented Strains

We efficiently disrupted the Nc*ROP40* gene in the Nc-Spain7 isolate by the insertion of a pyrimethamine resistance cassette (DHFR–TS), as was previously carried out for NcΔ*GRA7* [39]. PCR and sequencing confirmed the deletion of the Nc*ROP40* gene and the correct integration of DHFR–TS (Figure 2A).

In addition, the absence of expression of the NcGRA7 and NcROP40 proteins in their corresponding KO was verified by Western blot and immunofluorescence (Figure 2B,C). Anti-NcGRA7 and anti-ROP40 rabbit sera did not recognize any band in the KO strains, while the expected bands for NcGRA7 (33 and 18 kDa) and NcROP40 (53 kDa) were detected in the WT strain (Figure 2B). Similar results were observed by immunofluorescence analyses, and both proteins were only detected in the WT parasites, either secreted in the parasitophorous vacuole (NcGRA7) or located at the apical end of tachyzoites (NcROP40) (Figure 2C).

To verify the role of NcGRA7 and NcROP40 in the virulence of the parasite in the murine model, complemented strains were constructed by reintroduction of the target genes (Nc*GRA7* or Nc*ROP40*) in the corresponding KO parasites. The parasites were selected by FUDR treatment and checked by PCR to confirm efficient transfection of Nc*GRA7* or Nc*ROP40* (Figure 2A). Western blot and immunofluorescence analyses demonstrated the expression of the NcGRA7 and NcROP40 proteins in the complemented strains, with levels of expression comparable to those observed in the WT strain (Figure 2B,C).

The above results demonstrate that the *N. caninum*-deficient strains (NcΔ*GRA7* and NcΔ*ROP40*) and the complemented strains (iNcΔ*GRA7* and iNcΔ*ROP40*) were successfully generated.

### 3.2. Evaluation of NcGRA7 and NcROP40 Knockout Parasite Virulence in the BALB/c Model for Congenital and Cerebral Neosporosis

Data for pregnancy rates, litter size, mortality rates and parasite presence in dams are summarized in Table 2. No significant differences were found among the four groups in terms of pregnancy rate (*p* > 0.05, Fisher’s F test) or litter size (*p* > 0.05, one-way ANOVA). Pups born to dams infected with either KO parasites significantly prolonged their median survival times compared to the WT group (*p* < 0.05, log-rank test; Figure 3). The longest survival time was observed in the offspring from dams infected with the NcΔ*GRA7* parasites (15 days), followed by the pups from dams infected with NcΔ*ROP40* parasites (12 days) and finally, pups from the WT group (10 days), although all infected groups reached mortality rates close to 100% (Table 2). The complementation of NcΔ*GRA7* and NcΔ*ROP40* parasites resulted in a restoration of the parasite virulence, as offspring born to dams infected with the complemented parasites had the same median survival time as those from the WT group, with no significant difference (*p* = 0.54, log-rank test).

In dams, clinical signs were observed on the second week pi. Rough hair coats and apathy were the first signs observed, followed by anorexia, inactivity and nervous signs. In general, clinical signs of neosporosis were milder in the groups infected with the NcΔ*GRA7* or NcΔ*ROP40* parasites, although significant differences were only found between the NcΔ*GRA7* and WT groups (*p* < 0.05, Kruskal–Wallis, Dunn’s comparison post-test; Figure 4). Moreover, dams challenged with KO parasites had lower mortality rates than those infected with WT parasites (Table 2). Nevertheless, these differences were nonsignificant (*p* > 0.05, Fisher’s F test). In terms of the detection and quantification of *N. caninum* DNA, dams infected with the KO strains showed lower parasite burdens in the brain than dams from the WT group, but significant differences were found only with those infected with the NcΔ*GRA7* parasite (*p* < 0.05, Kruskal–Wallis, Dunn’s comparison post-test; Figure 5A). Dams with higher parasite loads also presented more severe clinical signs, with a significant positive correlation between parasite loads in the brain and clinical sign scores (*p* < 0.0001, Pearson r = 0.61 r^2^ = 0.37). Although nonpregnant mice had a lower parasite burden than the dams, lower parasite burdens were also observed in the groups infected with KO parasites (NcΔ*ROP40* or NcΔ*GRA7*) compared to the WT group, but without significant differences (*p* > 0.05, Kruskal–Wallis; Figure 5B).

### 3.3. Similar Cytokine Response Induced by NcGRA7 and NcROP40 Knockout and Wild-Type Parasites during the Acute Phase of Infection

Cytokine expression levels were measured in the spleen of males sacrificed at 5 days pi. All infected groups showed similar patterns, with higher expression levels of IFN-γ and IL-10 and lower expression levels of TNF-α and IL-4 (Figure 6). In addition, no significant differences were found between challenged groups for any cytokine (*p* > 0.05, Kruskal–Wallis), but higher expression levels of IFN-γ and IL-10 were observed in males infected with KO parasites compared to WT-infected males. The levels of IFN-γ and IL-10 were similar in each infected group, with IFN-γ/IL-10 ratios close to 1.

### 3.4. Humoral Immune Response Induced by NcGRA7 and NcROP40 Knockout and Wild-Type Parasites

Significantly higher levels of specific anti-*N. caninum* IgG1 and IgG2a antibodies were detected in all infected groups (dams and nonpregnant mice) compared to the negative control group (*p* < 0.0001, one-way ANOVA Tukey’s comparison post-test), confirming *N. caninum* infection. Dams infected with the NcΔ*GRA7* and NcΔ*ROP40* parasites had significantly reduced IgG1 levels compared to WT (*p* < 0.005, one-way ANOVA Tukey’s comparison post-test; Figure 7A). In nonpregnant mice, only those infected with NcΔ*GRA7* parasites had lower levels of IgG1 compared to the WT group (*p* < 0.001, one-way ANOVA Tukey’s comparison post-test; Figure 7B). No significant differences in IgG2a levels were observed between infected groups, either in dams or nonpregnant mice (*p* > 0.05, one-way ANOVA). Further analysis comparing IgG1/IgG2a ratios showed similar values between KO- and WT-infected groups.

## 4. Discussion

The pathogenesis of apicomplexan parasites is largely dependent on host–parasite interactions [23,48]. These parasites use specialized organelles, such as rhoptries and dense granules, to deliver proteins in their surrounding environment that, in many cases, are able to modulate the host response, allowing parasite survival and proliferation [26,49]. In *T. gondii*, many of these effector proteins have been described as virulence factors [24,50,51,52]. Although the understanding of the pathogenic molecular mechanisms in *N. caninum* is not as extensive as in the closely related *T. gondii*, some rhoptry (ROP16, ROP5) and dense granule (GRA7, GRA17, GRA6, GRA2) proteins have already been described as virulence factors in *N. caninum* [31,35,36,53,54,55].

Based on transcriptomic and proteomic comparisons between the isolates of different virulence carried out by our group [17,18,19,20,21], we hypothesize that NcGRA7 and NcROP40 proteins could contribute to parasite virulence. In addition, both proteins have been tested as putative vaccine candidates in mice, showing partial protection against infection [43,56,57,58].

Gene editing and deletion techniques are useful tools for studying protein functions in parasites and for better understanding host–parasite interactions. The CRISPR/Cas9 technology was used in *N. caninum* for the first time in 2018 to generate *N. caninum Gra7* defective parasites, demonstrating that the same constructs developed for the closely related *T. gondii* can be employed in *N. caninum* [39]. In this study, the NcΔ*ROP40* strain was generated following the same approach that was used for the generation of the Nc*GRA7*-deficient strain [39]. In contrast to other studies that failed or had very low efficiency disrupting genes in *N. caninum* using the tools developed for *T. gondii* [53,54,55], we successfully generated the KO and complemented lines using this system. In addition, we used the highly virulent and well-characterised isolate Nc-Spain 7 instead of Nc-1.

The importance of these proteins in parasite virulence was assayed in the well-established murine models of neosporosis. Although the role of the NcGRA7 protein in virulence has already been studied in mice, the models used have been poorly normalized. It has been widely shown that results in these models are dependent on the mouse breed, parasite isolate, infection dose, route of administration, and timing of infection during pregnancy. This is why previous studies have focused on standardizing and refining a pregnant BALB/c mouse model that could be used in different research groups with similar outcomes [37,38]. This model is sensitive enough to detect differences in virulence between *N. caninum* isolates and can be used to classify them as high, low, or moderate virulence [6,7]. In infections at midgestation with 10^5^ tachyzoites of Nc-Spain7, neonatal mortality and vertical transmission rates are expected to be close to 100%, with a high proportion of dams often developing severe clinical signs and mortality rates of approximately 40% [38].

The deletion of both Nc*GRA7* and Nc*ROP40* genes resulted in a reduced virulence in pups born from dams infected with the respective KO parasites, as their median survival times were significantly higher compared to that observed in the offspring from dams challenged with the WT parasite. These differences were not observed between the pups born to dams infected with the complemented strains and those from the WT group, supporting the role of NcGRA7 and NcROP40 in parasite virulence. The longest median survival time was recorded in the group infected with NcΔ*GRA7* parasites. However, offspring survival rates were similar in all infected groups, less than 5%, and almost all infected pups died. Vertical transmission was not analysed, but according to previous findings, the brains from pups that succumbed to infection were assumed to be *N. caninum* PCR-positive [5,38,43], suggesting vertical transmission close to 100%. The virulence reduction of these KO parasites was also observed in dams infected with these lines, with lower mortality rates and milder clinical signs. This largely reflects what was observed in the pups, with the group infected with the NcΔ*GRA7* having the mildest clinical signs and the lowest mortality rate.

Parasite load has proven to be a good indicator of parasite pathogenicity, with a positive correlation between the presence of clinical signs and severity of lesions and parasite burdens in the brain [59,60,61]. In this study, we also observed such a correlation. Dams infected with NcΔ*GRA7* had a significantly lower parasite burden than those infected with the WT. Additionally, in nonpregnant mice, we observed a lower parasite burden in the groups infected with the deficient parasites, but without significant differences. This is probably linked to the fact that pregnant mice are more susceptible to infection, presumably due to the regulation of immunity during pregnancy [57,62,63], and differences in virulence between isolates are easier to observe [6,43,57,64,65], corroborating that it is a more sensitive model than that of nonpregnant mice.

Previous studies have also reported lower mortality rates and lower parasite loads in the brains of mice infected with the Nc*GRA7*-deficient strain than in those infected with the parental Nc-1 strain [31,32]. In contrast, Abdou et al. [33] reported higher mortality rates and similar parasite burdens in the brains of dams infected with the Nc*GRA7*-deficient strain and its parental strain Nc-1 isolate. Nevertheless, they also observed decreased virulence in pups born to dams infected with an Nc*GRA7*-deficient parasite, with a lower vertical transmission of the KO parasite (70%) and a higher offspring survival rate (50%). The differences observed in our work could be due to the time of infection (3.5 days vs. 7 days of pregnancy), the strain used for the KO generation (Nc-1 vs. Nc-Spain7) and the mouse breed (C57BL/6 vs. BALB/c); thus, comparisons between studies should be made with caution. Infection at midgestation (Day 7) leads to higher morbidity, mortality, and vertical transmission rates in pups than infection at early or late gestation, while infection at late gestation produced the lowest vertical transmission [37]. Differences among pregnancy periods could be a consequence of the degree of placenta development and embryo implantation. At early gestation, *N. caninum* may not reach the foetus because implantation has not yet occurred. At midgestation the embryo is already implanted (day 4.5) and the placenta is not completely developed and, at late gestation the placenta is completely developed acting as a barrier [37]. This could explain the higher mortality rates observed in our work. Moreover, unlike the high-virulent isolate Nc-Spain7, Nc-1 is an isolate with an uncontrolled high number of passages in cell culture and prolonged cell culture maintenance can attenuate virulence of *N. caninum* in vivo [66]. Nc-1 isolate is classified as moderate-virulence, but the results across studies are inconsistent with survival rates of infected mice ranging from near 100% [33,60] to below 40% [31,32]. All these differences demonstrate the importance of using a well-established model and a characterized isolate that provides reproducible results.

To date, no studies have been conducted on Nc*ROP40*-deficient parasites, but other rhoptries, such as NcROP5 and NcROP16, have been shown to play an important role in virulence in mice, and parasites deficient in these proteins also displayed lower parasite loads in the brain [35,36]. The lower virulence of groups infected with the deficient strains may be due to a defect in some critical step of the lytic cycle or to a different regulation of immunity. It has been reported that NcGRA7 and NcROP40 may play important roles during tachyzoite invasion [29,34]. Nc*ROP40* mRNA levels are higher during the invasion phase (6 h pi) and during the egress-reinvasion phase (56 h pi) [34] and both proteins are more abundant in isolates with higher invasion rates at 8 h pi [4,5,17,19,21]. Moreover, studies using Nc*GRA7*-deficient and -nondeficient parasite lines have shown an altered modulation of the immune response [31,32], which could contribute to parasite survival, such as the higher clearance of the Nc*GRA7*-deficient parasites by macrophages compared to the parental strain [32]. In *T. gondii*, this protein forms a complex with ROP5/ROP18 and regulates the ROP18-specific inactivation of the immunity-related GTPase Irga6 [67,68]. In *N. caninum,* Wang et al. [32] also reported that infection with NcΔ*GRA7* parasites resulted in a higher recruitment of Irga6 to the parasitophorous vacuole membrane. For ROP40, studies in *T. gondii* using a Tg*ROP40*-deficient strain showed no differences in growth in vitro and virulence in mice compared to the parental strain [69] but exhibited a reduced cyst burden in the brain [70]. TgROP40 is differentially expressed during the biological cycle, with higher expression during the tachyzoite and sporulated oocyst stage, and may play different roles during different life cycle stages [69]. However, there are no similar studies on *N. caninum* and the molecular role of ROP40 remains unclear.

In terms of the humoral immune response, pregnant mice infected with KO parasites had significantly lower IgG1 levels than those challenged with WT parasites. The predominance of IgG1 is usually associated with a Th2 response [71], which likely contributes to a more efficient multiplication of the parasite in host tissues [62,72]. This is in line with the results that we obtained, where dams infected with the NcΔ*GRA7* strain had the lowest IgG1 levels and the greatest reduction in virulence, while WT-infected dams had the highest IgG1 levels and parasite loads. However, the cytokine production did not appear to result in a strong modulation of the immune response, as no significant differences were observed between infected groups, although certain patterns were discernible. The Th1-type immune response, characterized by the production of proinflammatory cytokines such as IFN- γ and IL-12, is essential for limiting parasite multiplication [61,73,74,75]. In our study, mice infected with KO parasites had higher IFN- γ levels, with the group infected with the NcΔ*ROP40* parasite having the highest production. The levels of IL-10 in the infected groups were similar to those of IFN- γ, with a balanced Th1/Th2 response. Abdou et al. [33] also reported similar expression levels of cytokines in the spleens of animals infected with Nc*GRA7*-deficient parasites and WT parasites, but other studies found lower IFN- γ levels in mice infected with parasites deficient in Nc*GRA7* than in mice infected with the parental strain Nc-1 [31,32]. Further research in this field is needed to elucidate the role of these proteins in the immune response and in cytokine production.

In the present study, we demonstrated that NcGRA7 is a virulence factor in *N. caninum* using a well-established pregnant BALB/c mouse model. In addition, we described NcROP40 as a new virulence factor in *N. caninum.* The lack of these proteins may cause defects in the invasion and replication capacity of mutant parasites, in the ability to cross barriers such as the placenta and the blood-brain barrier, or in the immune response evasion/stimulation mechanisms. All these questions provide new avenues for research. Further studies are needed to clarify the function of these proteins during *N. caninum* infection.

## Figures and Tables

**Figure 1 pathogens-11-00998-f001:**
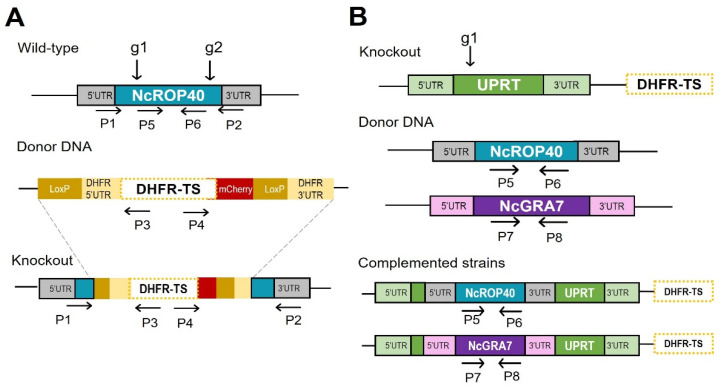
Construction and verification of the knockout and complemented strains by CRISPR/Cas9. (**A**) Schematic diagram of *NcROP40* gene disruption and the repair template in Nc-Spain7 (wild-type). Disruption was achieved by employing two guide RNAs (g1 and g2). The positions of primers used for diagnostic PCR (P1–6) are indicated by arrows. (**B**) Schematic diagram of *NcROP40* and *NcGRA7* complementation of the knockout parasites by insertion at the *UPRT* locus. The *UPRT* gene was disrupted employing one guide RNA (g1). The positions of primers used for diagnostic PCR (P5–8) are indicated by arrows. UTR, untranslated region; DHFR–TS, dihydrofolate reductase–thymidylate synthase; LoxP: loxP sites.

**Figure 2 pathogens-11-00998-f002:**
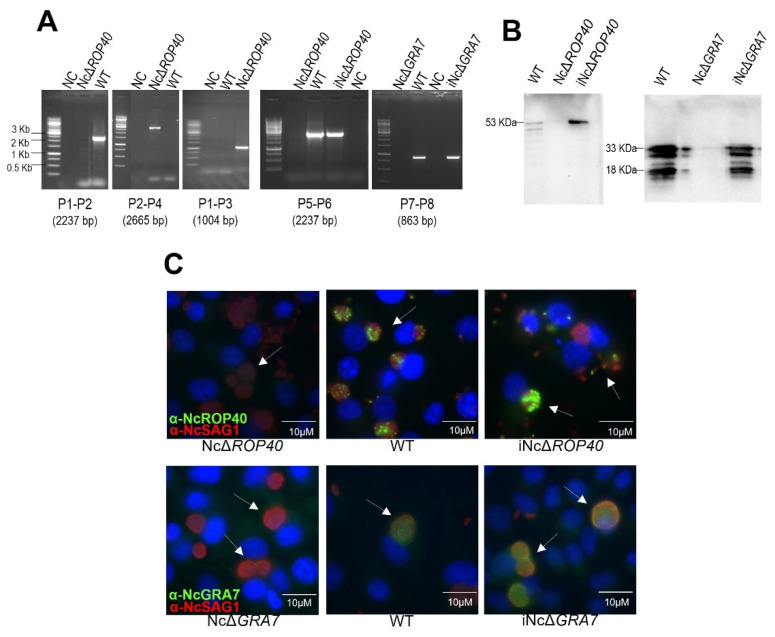
Verification of the knockout and complemented strains. (**A**) Diagnostic PCR demonstrating the loss of *NcROP40* (P1–P2, P5–P6) and the integration and orientation of the donor template (P2–P4, P1–P3) into the correct locus in the Nc*ROP40* knockout parasite. Diagnostic PCR demonstrating the integration of *NcROP40* (P5–P6) and *NcGRA7* (P7–P8) in the complemented strains. Primers used for PCR are described in Table 1 and indicated in Figure 1. (**B**) Western blotting assessment of *NcROP40* and *NcGRA7* gene disruption and complementation by measuring the protein expression levels. Anti-NcROP40 rabbit serum detected a 53-kDa protein in the WT and complemented strain (iNc∆*ROP40*) but not in the knockout (Nc∆*ROP40*) strain. A rabbit anti-NcGRA7 antibody detected two major proteins of 33 and 18 kDa in the WT and in the complemented strain (iNc∆*GRA7*), but not in the knockout parasite (Nc∆*GRA7*). (**C**) Immunofluorescence staining of Marc-145 cells infected with different strains of *Neospora caninum* (WT, knockouts, and complemented strains). The nuclei were stained with DAPI (blue), and the monoclonal antibody α-NcSAG1 was used as a control to mark the periphery of the parasites (red). The images above show methanol-fixed cultures labelled with affinity purified polyclonal antibody α-NcROP40 (green) and the images below show paraformaldehyde/glutaraldehyde-fixed cultures labelled with affinity purified polyclonal antibody α-NcGRA7 (green). White arrows indicate the expression or lack of expression of the proteins NcROP40 or NcGRA7 into the parasitophorous vacuole. Note that NcROP40 and NcGRA7 protein are detected in the WT and in the complemented strains but not in the knockout strains. WT, wild type (NcSpain7); NC, negative control; NcΔ*ROP40*, Nc*ROP40* knockout parasite; NcΔ*GRA7*, Nc*GRA7* knockout parasite; iNcΔ*ROP40*, complemented strain of Nc*ROP40* knockout parasite; iNcΔ*GRA7*, complemented strain of Nc*GRA7* knockout parasite.

**Figure 3 pathogens-11-00998-f003:**
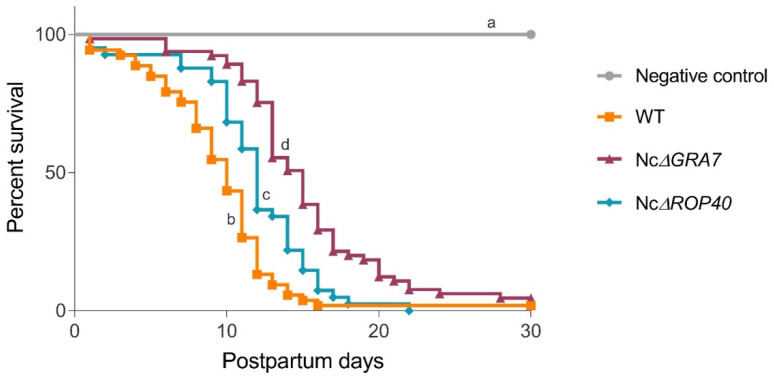
Effect of *Neospora caninum* infection in the offspring during a period of 30 days pp (postpartum). Kaplan–Meier survival curves of pups from dams infected on Days 7 of pregnancy with 10^5^ tachyzoites from different *N. caninum* strains and the uninfected group (negative control, inoculated with PBS). The wild-type (WT) group was challenged with the Nc-Spain7 isolate, and the NcΔ*GRA7* and NcΔ*ROP40* groups were infected with knockout mutant parasites (deficient in GRA7 and ROP40 proteins, respectively). Each point represents the percentage of surviving animals on that day, and vertical steps downwards correspond to observed deaths. Letters a, b, c and d indicate significant differences (*p* < 0.05, log-rank test). Note that a delay in time to death was observed in groups infected with mutant parasites compared to the WT group.

**Figure 4 pathogens-11-00998-f004:**
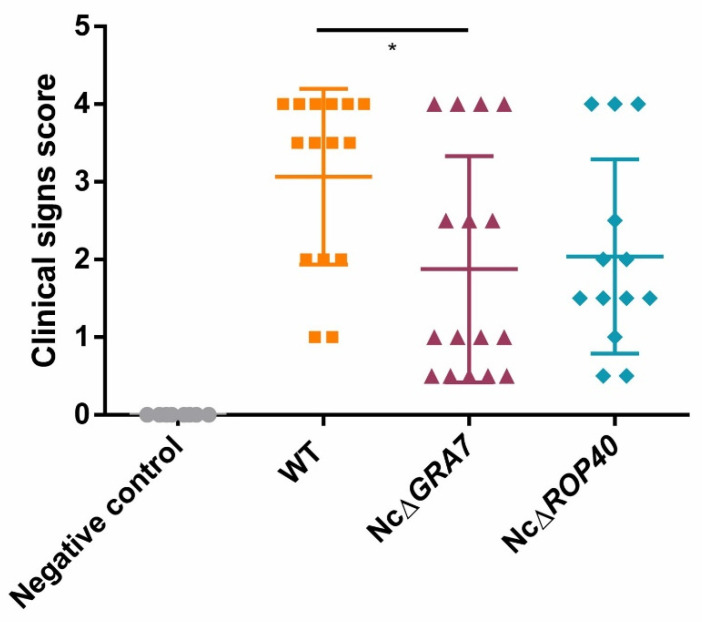
Clinical signs of dams infected with *Neospora caninum* tachyzoites (10^5^ tachyzoites/mouse). The wild-type (WT) group was infected with the Nc-Spain7 isolate, and the NcΔ*GRA7* and NcΔ*ROP40* groups were infected with knockout mutant parasites (deficient in GRA7 and ROP40 proteins, respectively). The negative control was challenged with PBS. Scores were based on the detection and severity of clinical signs (0, no alterations; 1, ruffed coat; 2, rounded back; 3, severe weight loss or 4, nervous signs). Each point represents a single animal. Significant differences between infected groups are denoted by horizontal lines and asterisks (* *p* < 0.05; Kruskal–Wallis Dunn’s comparison post-test).

**Figure 5 pathogens-11-00998-f005:**
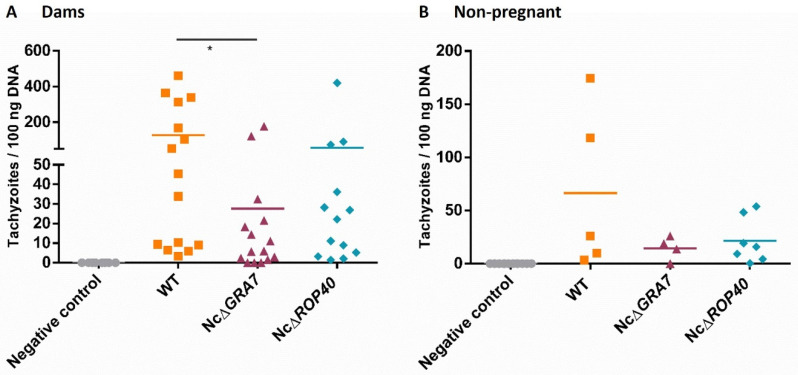
Cerebral parasite burden at 30 Days postinfection in mice infected with 10^5^ tachyzoites of different *Neospora caninum* strains and the uninfected group (negative control). The wild-type (WT) group was challenged with the Nc-Spain7 isolate, and the NcΔ*GRA7* and NcΔ*ROP40* groups were infected with knockout mutant parasites (deficient in GRA7 and ROP40 proteins, respectively). Each dot represents individual values, and medians are represented as horizontal lines. Parasite burden in the brain was calculated using real-time qPCR. (**A**) Parasite burden in the brains of dams infected on Day 7 of gestation. (**B**) Parasite burden in the brains of nonpregnant mice. Significant differences between infected groups are denoted by horizontal lines and asterisks (* *p* < 0.05, Kruskal–Wallis Dunn’s comparison post-test).

**Figure 6 pathogens-11-00998-f006:**
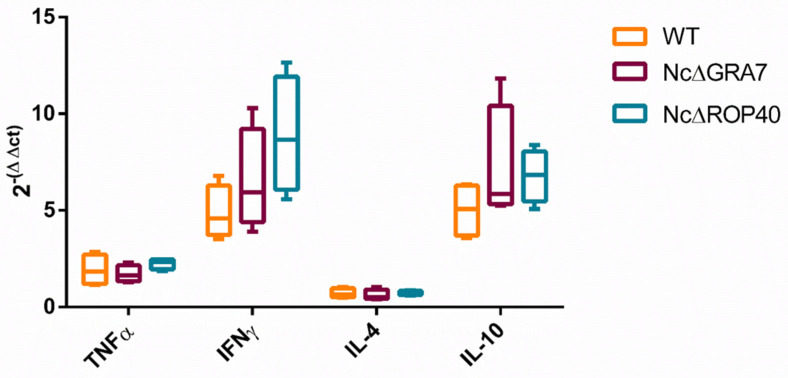
Cellular immune response in *Neospora caninum*-infected mice during the acute phase (5 days postinfection). The wild-type (WT) group was challenged with the Nc-Spain7 isolate, and the NcΔ*GRA7* and NcΔ*ROP40* groups were infected with knockout mutant parasites (deficient in GRA7 and ROP40 proteins, respectively). Box-plot graphs represent the cytokine expression in the spleen, the mean (horizontal lines), the lower and upper quartiles (boxes) and minimum and maximum values (whiskers). Each sample was normalized to β-actin expression and relative to the negative control group. The results are given by 2^−∆∆Ct^, and the comparative threshold cycle method was used. According to the Kruskal–Wallis test, no significant differences were present between infected groups.

**Figure 7 pathogens-11-00998-f007:**
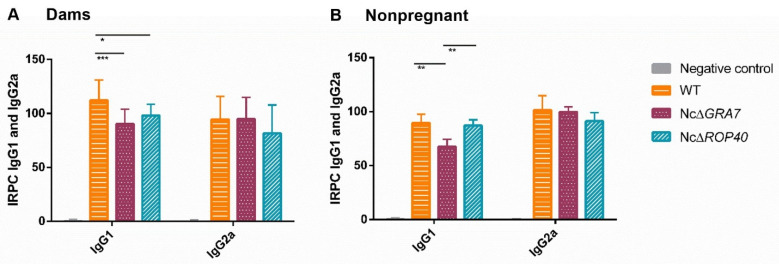
Humoral immune responses in challenged mice at 30 days postinfection. Anti-*N. caninum* immunoglobulins (IgG1 and IgG2a isotypes) generated in (**A**) dams and (**B**) nonpregnant mice inoculated with 10^5^ tachyzoites from different *N. caninum* strains and the uninfected group (PBS). The wild-type (WT) group was challenged with the Nc-Spain7 isolate, and the NcΔ*GRA7* and NcΔ*ROP40* groups were infected with knockout mutant parasites (deficient in GRA7 and ROP40 proteins, respectively). Bars represent the average RIPC (relative index percent), and error bars represent standard deviations for each group. Significant differences between infected groups are denoted by horizontal lines and asterisks with horizontal lines (*** *p* < 0.001; ** *p* < 0.01; * *p* < 0.05, one-way ANOVA Tukey’s comparison post-test).

**Table 1 pathogens-11-00998-t001:** Guide RNAs (gRNAs) and primers used to generate the knockout parasites and complemented strains.

Purpose	Primer Name	Sequence	Used to
**Nc*ROP40*** **disruption**	ROP40 gRNA1	5′ GATATTTGCACTCGAATGCT 3′	Disrupt ROP40 gene in 5′ and 3′ terminal ends
	ROP40 gRNA2	5′ GTCGCGGTGCGTTCAGTGGTG 3′	
	ROP40 Fw (P1)	5′ TAAGAACGCATGGCTGACTG 3′	Amplify and sequence the gRNA targeted region of the ROP40 gene together with donor template
	ROP40 Rv (P2)	5′ CGGTTCGGACAAAACGTATAC 3′	
	DHFR Fw (P4)	5′ GGCGTGAAGATCTGGGACAA 3′	Amplify and sequence the gRNA targeted region of the ROP40 gene together with DHFR–TS donor template
	DHFR Rv (P3)	5′ GCCTGGTATCTTTATAGTCC 3′	
**Nc*ROP40* and Nc*GRA7* ** **complementation**	UPRT gRNA	5′ GCAGGAGGAAAGCATTCTGC 3′	Disrupt UPRT gene in 5′ terminal end
	*KpnI*_ROP40 Fw	5′ AGTCGAggtaccGAGTGCATGAGGGAGTTCAAG 3′	Amplify the ROP40 gene with its promoters for cloning it into the pUC19-plasmid; the restriction sequence is shown in lower case
	*KpnI*_ROP40 Rv	5′ TCGACTggtaccGCAGTCAGAACCACGTTTTCC 3′	
	iROP40 Fw (P5)	5′ TAAGAACGCATGGCTGACTG 3′	Amplify and sequence the ROP40 complemented gene donor template
	iROP40 Rv2 (P6)	5′ CGGTTCGGACAAAACGTATA 3′	
	*KpnI*_GRA7 Fw	5′ AGTCGAggtaccAAACACAGGTTCGTTCCTGCC 3′	Amplify the GRA7 gene with its promoters for cloning it into the pUC19-plasmid; the restriction sequence is shown in lower case
	*KpnI*_GRA7 Rv	5′ TCGACTggtaccTCGAAGCAGAGAGAAGCTTC 3′	
	iGRA7 5UTR Fw (P7)	5′ TCGCTGTTCCTGTAGGCTTT 3′	Amplify and sequence the GRA7 complemented gene donor template
	iGRA7 3UTR Rv (P8)	5′ CTGTCATCTGGGACACGAAA 3′	

**Table 2 pathogens-11-00998-t002:** Effect of *Neospora caninum* infections in BALB/c pregnant dams and their pups. Mice were subcutaneously infected with 10^5^ tachyzoites of the wild type strain Nc-Spain 7 (WT) or with the Knock-out strains: Nc*GRA7* deficient line (NcΔ*GRA7*) and Nc*ROP40* deficient line (NcΔ*ROP40*). One group was inoculated with PBS (Negative control).

Group	Fertility Rate ^a^	Litter Size ^b^	Mortality of Dams ^c^	Parasite Presence in Dams’ Brains ^d^	Early Pup Mortality ^e^	Neonatal Mortality ^f^	Median Survival Time of Pups ^g^
Negative control	8/20	5.38 ± 2.13	0/8	0/8	2/42	0/40	Undefined
	(40%)		(0%)	(0%)	(4.8%)	(0%)	
WT	15/20	4.64 ± 2.10	7/15	15/15	14/65	50/51	10 days
	(75%)		(46.6%)	(100%)	(21.5%)	(98%)	
NcΔ*GRA7*	16/20	4.69 ± 1.74	4/16	13/16	11/75	61/64	15 days
	(80%)		(25%)	(81.25%)	(14.7%)	(95.3%)	
NcΔ*ROP40*	13/20	3.54 ± 1.61	3/13	13/13	8/46	38/38	12 days
	(65%)		(23.1%)	(100%)	(17.4%)	(100%)	

^a^ Number of pregnant mice/number of mice mated in the group (percentage). ^b^ Average ± SD. ^c^ Number of dead or culled dams before 30 d pp/number of dams in the group (percentage). ^d^ Number of brain PCR-positive dams/number of dams in the group (percentage). ^e^ Number of stillborn and dead pups from birth to day 2/number of born pups in the group (percentage). ^f^ Number of dead pups from day 2 until the end of the experiment/number of pups alive by day 2 pp (percentage). ^g^ Day pp at which 50% of pups succumbed to infection.

## Data Availability

Data supporting the conclusions of this article have been included within the article and its additional files.

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
