# Peer review of "NcGRA7 and NcROP40 Play a Role in the Virulence of Neospora caninum in a Pregnant Mouse Model"

_pathogens, 2022, doi:10.3390/pathogens11090998_

Round 1

Reviewer 1 Report

NcGRA7and NcROP40 play a role in the virulence of Neospora caninum in a pregnant mouse model- pathogens-1865666.  

General comments 

I found the article remarkably interesting and easy to read. Minor modifications are needed to improve the Ms.  

Introduction  

Line 32: Neospora caninum is an obligate intracellular apicomplexa parasite, phylogenetically related to Toxoplasma gondii, and the aetiological agent of bovine neosporosis. Add the comma.  

Line 35: Please modify this sentence: Unfortunately, no tools are available.  

Line 44: Think to delete: showed.  

Line 40: Delete observed.  

Line 57: and host response control...Please add a reference/s.  

Please, include the studies in other rhoptries to discuss later.  

Materials and Methods 

Line 87: Please give details of the Figure or supplementary material.  

Fig 1 E: Please add scale bar to all images. If possible, indicate with a narrow  

Line 245-246: Could be possible to move this to 2.3 please? 

Results 

I prefer to see figures 1 C, D AND E,later in the Ms. I suggest creating a different Figure. 

Fig 1 C: Please add ladder details and bp length reference of the bands. 

3.3. Line 442. Why only in males? Please clarify this. 

Discussion  

Line 442: Please add some references.  

Line 511 – 514: these sentences go to M&M. 

Line 562-562: How do you explain this? Any hypothesis? It would be interesting to discuss this more.  

Line 565-568: Please rewrite this. It is not clear.  

Line 569-585: Is there any transcriptomic data of these genes between different stages during the biological cycle?

Reviewer 2 Report

Dear authors,

You did an excellent and very hard job trying to unveil another little piece of the Neospora c. pathogenesis. A great thumb up for all the group.

Here are some very minor suggestions, which I think it might improve the manuscript:

Line 13: "biological variability" is a generic term which can be interpreted in broad array of meanings. Can you find a more precise and less ambiguous term for it?

Line 19: Instead of “partial loss of virulence” I would say “reduction of virulence”. It is clearer to me.

Line 32: ApicomplexaN

Line 33: “Etiological” should be if you adopted American English. Decide if you want to adopt British or American and check thoroughly the entire manuscript.

Lines 261-262: when you are defining the four groups, I had the impression that you just mention three (because you just wrote the “KO parasites”). I would rephrase it giving the readers, 4 separate groups, without brackets.

Line 323: You mention that fertility and mortality rates are organised in contingency table. Can you be clearer about it?

Line 327: You rightfully describe your statistical methods for analysis, with ANOVA and Kruskal-Wallis for different values. I assume that you first verified the distribution type. Can you add which tests you used to verify the distribution?

Lines 368-369: “…prolonged the their median survival times.” Here you should mention compared to who (I know you explain in detail afterwards, but the sentence seems cut).

Lines 447-448: “The ratio of IFN…..” The whole sentence should go into M&M.
